

# Identification of aluminum-activated malate transporters (ALMT) family genes in hydrangea and functional characterization of *HmALMT5/9/11* under aluminum stress

Ziyi Qin[1,2,*], Shuangshuang Chen[2,*], Jing Feng[2], Huijie Chen[2], Xiangyu Qi[2], Huadi Wang[2,3] and Yanming Deng[1,2,3]

[1] College of Horticulture, Nanjing Agricultural University, Nanjing, Jiangsu, China
[2] Jiangsu Key Laboratory for Horticultural Crop Genetic Improvement, Institute of Leisure Agriculture, Jiangsu Academy of Agricultural Sciences, Nanjing, Jiangsu, China
[3] School of Life Sciences, Jiangsu University, Zhenjiang, Jiangsu, China
* These authors contributed equally to this work.

Corresponding author
Yanming Deng, dengym@jaas.ac.cn

## ABSTRACT

Hydrangea (*Hydrangea macrophylla* (Thunb.) Ser.) is a famous ornamental plant species with high resistance to aluminum (Al). The aluminum-activated malate transporter (ALMT) family encodes anion channels, which participate in many physiological processes, such as Al tolerance, pH regulation, stomatal movement, and mineral nutrition. However, systematic studies on the gene family have not been reported in hydrangea. In this study, 11 candidate ALMT family members were identified from the transcriptome data for hydrangea, which could be divided into three clusters according to the phylogenetic tree. The protein physicochemical properties, phylogeny, conserved motifs and protein structure were analyzed. The distribution of base conservative motifs of HmALMTs was consistent with that of other species, with a highly conserved WEP motif. Furthermore, tissue-specific analysis showed that most of the *HmALMTs* were highly expressed in the stem under Al treatment. In addition, overexpression of *HmALMT5*, *HmALMT9* and *HmALMT11* in yeasts enhanced their tolerance to Al stress. Therefore, the above results reveal the functional role of *HmALMTs* underlying the Al tolerance of hydrangea. The present study provides a reference for further research to elucidate the functional mechanism and expression regulation of the *ALMT* gene family in hydrangea.

## INTRODUCTION

Approximately 30% of the world's land is acidic soil, and as much as 50% of the cultivable land in the world is acidic (*von Uexküll & Mutert, 1995*). In acidic soil, aluminum (Al) is considered to be the main factor restricting plant growth and crop yield (*Kochian, Piñeros & Hoekenga, 2005*). Excessive Al in soil or substrate causes a series of negative effects on

plant nutrient uptake, enzyme activities, cell division and other physiological and biochemical processes, resulting in the inhibition of root growth and function, which may ultimately affect related processes (*Kochian, 1995*; *Sade et al., 2016*; *Sharma, Vasudeva & Kaur, 2006*). When Al exists in acidic soil in the form of soluble $Al^{3+}$, it has a toxic effect on plants. At present, supplying limestone and alkaline fertilizer is the most common method to resolve Al toxicity in acidic soil, but these two substances only solve the pollution problem of surface soil and cannot eradicate the problem of soil acidification. Additionally, the above methods may also cause potential environmental pollution. Therefore, cultivating Al-tolerant plants may be an effective way to avoid Al toxicity.

To grow in acidic soil, plants have evolved strategies to cope with Al toxicity. Numerous studies have shown that the Al tolerance mechanism of plants includes internal detoxification of Al (*Ma et al., 1997*), root secretion of phenolic compounds (*Chen et al., 2013*), root-mediated pH changes in the rhizome layer (*Wehr, Menzies & Blamey, 2003*), and cell wall pectin content and degree of methylation (*Eticha, Stass & Horst, 2005*). However, the processes leading to gene expression in secretion and during the Al response and tolerance are still unknown. In many Al-tolerant plants, the secretion of organic acids (such as malate, citrate and oxalate) was stimulated by Al in root tips, which was considered to be one of the most important mechanisms of plant resistance to Al (*Andrade et al., 2011*; *Hoekenga et al., 2003*; *Pellet, Grunes & Kochian, 1995*). In this case, anion channels were involved in the Al tolerance mechanism due to the efflux of $Al^{3+}$ chelating malate or citrate anions through these channels. Al-induced malate transporter and citrate transporter genes have been successfully cloned, and the secretion of malic acid and citric acid was controlled by aluminum-activated malate transporter (ALMT) and multidrug and toxic compound extrusion transporter (MATE), respectively (*Cardoso, Pinto & Paiva, 2020*; *Ryan et al., 2011*). The *ALMT* gene family is unique to plants. The family members encode transmembrane proteins, which are widespread in plants and participate in many physiological processes, such as Al tolerance, pH regulation, stomatal movement, and mineral nutrition (*Dreyer et al., 2012*; *Gruber et al., 2010*; *Ligaba et al., 2006*; *Ligaba et al., 2012*). For example, wheat *TaALMT1* is involved in the mechanism of organic acid secretion from roots to soil. In the past two decades, some ALMTs have been identified in different plants. The functions of most *ALMT* gene family members were related to Al tolerance, such as *ScALMT1*, which participates in the physiological process of Al tolerance in rye (*Collins et al., 2008*). In addition, *AtALMT1* isolated from *Arabidopsis thaliana* was an important factor in Al tolerance (*Hoekenga et al., 2006*), and *BnALMT1*, *BnALMT2* from rape encoded ALMTs enhancing the Al resistance of plant cells (*Ligaba et al., 2006*).

*Hydrangea macrophylla* is an important ornamental plant species in both parks and home gardens, with high ecological and economic value. Hydrangea is a well-known Al accumulating and tolerant plant, which can accumulate 5 mg·g$^{-1}$ $Al^{3+}$ (dry weight, DW) in leaves without symptoms of Al damage (*Ma et al., 1997*). Hydrangea flower sepals turn from red to blue by adding $Al^{3+}$ to acidic soil, and the content of $Al^{3+}$ in the blue sepals of hydrangea was approximately 40 times higher than that in red sepals (*Ito et al., 2009*; *Schreiber et al., 2011*). Studies have shown that *HmABCs*, *HmVALT* and other genes play important roles in the Al tolerance of hydrangea (*Chen et al., 2015*). However, the

mechanisms of the tolerance and enrichment of Al in the roots are still unclear. Therefore, the identification and analysis of related genes is important to reveal the molecular mechanism of Al tolerance in hydrangea.

The ALMT family has been identified and characterized in many species. For instance, 34 *GmALMT* members were identified in the soybean genome, and the identified genes were proven to improve the utilization of dilute soluble phosphorus (P) in roots by mediating malate secretion (*Dos et al., 2018*; *Peng et al., 2018*). In rubber tree, 17 members of the *ALMT* gene family were identified, and four of them were involved in Al detoxification (*Ma et al., 2020*). In addition, the *ALMT* gene family has been identified and analyzed in Chinese white pear and apple, and both play important roles in various physiological processes, such as malic acid accumulation and organic acid efflux (*Linlin et al., 2018*; *Ma et al., 2018*). However, the ALMT gene has not yet been systematically identified under Al stress in hydrangea. In the present study, 11 members of the *ALMT* gene family were identified and analyzed from hydrangea by bioinformatics. This study focused on the identification, phylogeny, evolution and structural analysis of the *ALMT* gene family members in hydrangea. In addition, the function of the genes in enhancing Al tolerance was verified in yeast. The results contribute to revealing the role of ALMTs in the mechanism of Al tolerance in hydrangea and provide evidence for future research on the evolution and genetic resources of ALMTs.

## MATERIALS AND METHODS

### Plant material and Al treatment

*H. macrophylla* 'Bailer' (Endless Summer™) was cultivated at the Preservation Centre of the Hydrangea Germplasm Resource at the Jiangsu Academy of Agricultural Sciences in Nanjing, China. All the plants were potted in plastic flowerpots with a diameter of 21 cm in growth chamber. The growth chamber was set as 25 ± 2 °C and 16 h light/day. During the squaring period, Hoagland solution (500 ml per pot) with or without 15 mM $Al_2(SO_4)_3$, was used to irrigate the plants once a week. The pH of Hoagland solution used in each treatment was 2.5 to 3.0, and each treatment contained 20 replicate pots. Root, stem, leaf and flower samples were collected when the flower color of 'Bailer' turned blue. All the samples were frozen in liquid nitrogen and stored at −80 °C. Three biological replicates were analyzed for each treatment.

### Identification of ALMTs in hydrangea

The ALMT homologous sequences in *Arabidopsis* were downloaded from TAIR (http://www.arabidopsis.org/) and used as query sequences to identify the candidate genes in the transcriptome data of 'Bailer' using the local BLAST program (e-value ≤ $1e^{-5}$) (*Chen et al., 2015*). A Hidden Markov model (HMM) of the lateral organ boundary domain (PF11744) was used as the seed model for the HMMER3 search (http://hmmer.janelia.org/) of the local 'Bailer' protein database ($E ≤ 10^{-20}$) (*Finn, Clements & Eddy, 2011*). Furthermore, the Pfam database (http://pfam.xfam.org/) and SMART programs (http://smart.embl-heidelberg.de/) were used to verify the candidate HmALMTs.

## Analysis of protein properties, conserved domains and motifs

The online ExPASy software (http://web.expasy.org/protparam/) was used to analyze the number of amino acids, isoelectric point and molecular weight of HmALMTs. The subcellular localization and transmembrane regions were predicted by Cell-Ploc 2.0 (http://www.csbio.sjtu.edu.cn/bioinf/Cell-PLoc-2/) and TMHMM (http://www.cbs.dtu.dk/services/TMHMM/), respectively. Analysis of the protein secondary structure was carried out with the SOPMA tool (https://npsa-prabi.ibcp.fr/cgi-bin/npsa_automat.pl?page=/NPSA/npsa_sopma.html), and the protein three-dimensional (3D) structure prediction was predicted by I-TASSER (https://zhanggroup.org//I-TASSER/) and viewed using PyMOL software (*Zhang, 2008*). MEME (https://meme-suite.org/meme/) was used to analyze protein conserved motifs with the following parameters: number of repetitions = any; maximum number of motifs = 25; and optimum motif width = 6 – 100 residues.

## Sequence alignments and phylogenetic analysis

The ALMT homologs in *Arabidopsis thaliana* (14 members) were obtained from TAIR (http://www.arabidopsis.org/), and those in *Apostasia shenzhenica* (seven members), *Erythranthe guttata* (18 members), *Nicotiana tabacum* (10 members) and *Rosa chinensis* (18 members) were obtained from NCBI (https://www.ncbi.nlm.nih.gov/). Multiple sequence alignments were executed in ClustalX with the default parameters (gap opening = 10; gap extension = 0.2; delay divergent sequences (%) = 30; DNA transition weight = 0.5; use negative matrix = off) (*Thompson, Gibson & Higgins, 2002*). Subsequently, MEGA7.0 software was used to construct a phylogenetic tree and analyze the molecular evolutionary ship with the neighbor-joining (NJ) method and the bootstrap test replicated 1,000 times (*Sudhir, Glen & Koichiro, 2016*).

## RNA isolation and real-time quantitative PCR (qRT–PCR) analysis

Total RNA was extracted using the Plant RNA Isolation Kit (YEASEN, Shanghai, China), and its quality and concentration were detected using a NanoDrop 2000 spectrophotometer. TranScript One-Step gDNA Removal RNase-free was used to digest the DNA of all samples. Then, cDNA Synthesis SuperMix (YEASEN, Shanghai, China) was used to synthesize the first-strand cDNA. qRT–PCR with SYBR Green I Master Mix (YEASEN, Shanghai, China) was performed on an Applied Biosystems 7500 Real-Time PCR System (Applied Biosystems, Foster City, CA, USA). The composition of the PCR mixture was as follows: 0.4 μl of each primer (10 μM), 10 μl of 2 × SYBR Green I Master Mix, 2 μl of cDNA, and 7.2 μl of RNase-free water. qRT–PCR began with 30 s at 95 °C, followed by 40 cycles of 95 °C for 5 s and 60 °C for 34 s; then, the melting curve stage was conducted at the instrument default setting. The relative expression level of each gene was measured according to the cycle threshold (Ct), also known as the $2^{-\Delta\Delta CT}$ method, and all the analyses consisted of three biological replicates. *UPL7* and *β-TUB* were selected as the reference genes (*Chen et al., 2021*). All the primer sequences (designed using Primer 3.0) used for qRT–PCR are listed in Table S1.

**Table 1 The basic information of the identified HmALMTs in hydrangea.**

| Name | Number of amino acids (aa) | MW (kDa) | pI | Instability index | Aliphatic index | GRAVY | TMD | Subcellular location |
|---|---|---|---|---|---|---|---|---|
| HmALMT1 | 475 | 52,803.12 | 8.52 | 29.88 | 97.12 | 0 | 6 | Chloroplast |
| HmALMT2 | 395 | 43,839.39 | 9.01 | 38.08 | 102.76 | 0.23 | 6 | Peroxisome |
| HmALMT3 | 550 | 61,729.03 | 6.48 | 37.84 | 94.33 | −0.057 | 6 | Cell membrane |
| HmALMT4 | 387 | 42,726.91 | 6.62 | 32.68 | 104.81 | 0.251 | 6 | Peroxisome |
| HmALMT5 | 438 | 48,277.61 | 8.15 | 34.14 | 103.95 | 0.245 | 6 | Peroxisome |
| HmALMT6 | 403 | 45,610.43 | 5.44 | 52.34 | 79.63 | −0.487 | 0 | Nucleus |
| HmALMT7 | 408 | 45,101.18 | 5.54 | 27.17 | 94.14 | 0.061 | 5 | Nucleus |
| HmALMT8 | 487 | 53,618.93 | 7.63 | 32.81 | 97.35 | 0.032 | 5 | Chloroplast |
| HmALMT9 | 314 | 34,906.71 | 8.07 | 23.27 | 101.91 | 0.329 | 6 | Cell membrane |
| HmALMT10 | 479 | 53,026.74 | 6.14 | 35.69 | 99.33 | 0.16 | 6 | Chloroplast |
| HmALMT11 | 529 | 59,104.55 | 9.1 | 36.03 | 98.83 | −0.014 | 6 | Chloroplast |

**Note:**
MW, molecular weight; pI, Isoelectric point; and TMD represents the number of transmembrane helices predicted by TMHMM online software, respectively.

## Heterologous expression of *HmALMTs* in yeast

The specific primers *HmALMT11*-F/R, *HmALMT9*-F/R, and *HmALMT5*-F/R (Table S1) were used to amplify the CDS in *H. macrophylla*. Then, the purified PCR products were subcloned into the *Kpn*I and *Not*I sites of the pYES2.0 vector to generate pYES-*HmALMT11*, pYES-*HmALMT9*, and pYES-*HmALMT5* fusion constructs. The lithium acetate method was used to transform the BY4741 strain (*Gietz & Woods, 1998*). The empty pYES2.0 vector was used as a control. To test the Al tolerance, those transgenic yeast lines were grown to $OD_{600} = 1$ and then serially diluted ($OD_{600} = 10^0$, $10^{-1}$, $10^{-2}$, $10^{-3}$, $10^{-4}$, and $10^{-5}$), spotted on Synthetic Galactose Minimal Medium without Uracil (SG-U) agar plates supplemented with 0, 1, and 2 mM $Al_2(SO_4)_3$ and incubated at 28 °C for 3 d. The pH of SG-U medium with or without $Al_2(SO_4)_3$ was adjusted to 4.0 ± 0.2. The relative growth of transformants was determined by measuring the $OD_{600}$ at 12 h intervals.

# RESULTS

## Identification and characterization of ALMTs in hydrangea

A total of 19 genes were filtered from the annotated file of the *Hydrangea* transcriptome database with the keywords 'ALMT' and 'aluminum-activated malate transporters'. After deredundancy (HMMER search and BLASTP program), SMART and Pfam identification, 11 *HmALMT* genes were identified. The amino acid sizes of HmALMTs ranged from 266 aa to 550 aa, their molecular weights were from 30,313.41 to 61,729.03 Da, and the theoretical isoelectric point (*pI*) varied from 5.44 to 9.10 (Table 1). TMHMM predicted that all identified HmALMTs except for HmALMT6 contained 5–6 transmembrane regions at the N-terminus of the ALMT protein. Furthermore, the HmALMTs were mainly located in the cytoplasm, with others located in the cell membrane, peroxisome and nucleus, as predicted by the subcellular location analysis (Table 1).

## Phylogenetic analysis of ALMTs in hydrangea

To explore the evolutionary relationship of ALMTs, the amino acid sequences from *Hydrangea* (11 members), *Arabidopsis thaliana* (14 members), *Apostasia shenzhenica* (seven members), *Erythranthe guttata* (18 members), *Nicotiana tabacum* (10 members) and *Rosa chinensis* (18 members) were selected to construct a phylogenetic tree. The above 78 ALMTs could be classified into three clusters. However, the 11 HmALMT members were distributed unevenly in three clusters, with five being in Cluster I (HmALMT1, HmALMT2, HmALMT7, HmALMT8 and HmALMT10), five in Cluster II (HmALMT3, HmALMT4, HmALMT5, HmALMT6 and HmALMT9), and only one (HmALMT11) in Cluster III (Fig. 1).

## Conserved domain and motif analysis of HmALMTs

The conserved domain analysis showed that the HmALMTs included ALMT (PF11744), FUSC_2 (PF13515) and FUSC (PF04632). A total of 15 conserved motifs were predicted and displayed a very diverse distribution pattern, validating their phylogenetic classification (Fig. 2, Table S2). The number of motifs involved in HmALMTs was also different. Motifs 1, 3 and 5 were found in all of the HmALMTs except for HmALMT6, and motifs 2, 4 and 8 were distributed in all of the HmALMTs except for HmALMT9. Motif 15 existed in only HmALMT6 and HmALMT8, and motif 12 was only discovered in HmALMT7 and HmALMT10 (Fig. 2).

## Analysis of the secondary and 3D protein models of HmALMTs

The secondary structure analysis showed that the protein encoded by the HmALMT family genes had α-helix, extended strand, β-turn and random coils, with mainly being α-helix and random coils. The proportions of the constituent elements of the secondary structure of all HmALMTs, ranking from high to low, were α-helix, random coil, extended strand and β-turn. Among them, with the exception of HmALMT6, which accounted for 48.88%, the α-helix of the other HmALMTs accounted for more than half of the constituent elements. For the extended strand, HmALMT9 had the highest proportion (15.29%), and other proportions ranged between 9–13%. Furthermore, the β-turns accounted for 2–4% (Table 2).

The prediction of the 3D protein model showed that all the HmALMT proteins had highly similar structures (Fig. 3). Except for HmALMT6, the 3D structures of other proteins were similar to 7VOJ (AtALMT1), and the C- and TM-scores were higher than −3 and 0.7, respectively (Table S3). According to the prediction of secondary structure and 3D protein models, all members of the ALMT family had a high structural similarity in the part near the N-terminal of the protein, whereas the part near the C-terminal diverged greatly.

## Expression of *HmALMTs* under Al stress

To understand the tissue expression patterns of the *ALMT* gene family in hydrangea, the roots, leaves, stems and flowers harvested from the control (Hoagland solution without $Al_2(SO_4)_3$) and treated plants (Hoagland solution with 15 mM $Al_2(SO_4)_3$) were used in

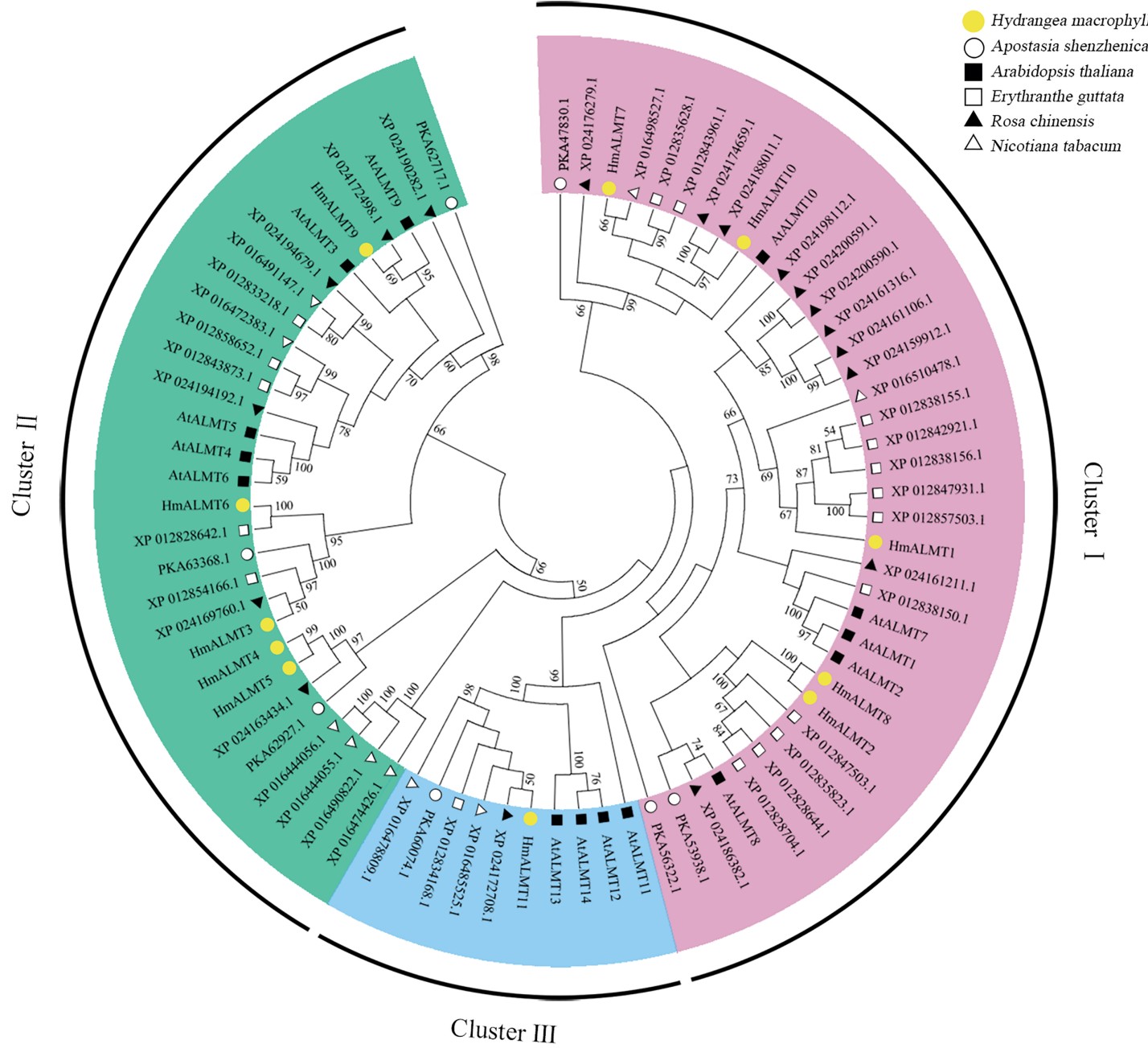

**Figure 1 Phylogenetic analysis of ALMTs from hydrangea, *Arabidopsis thaliana*, *Apostasia shenzhenica*, *Erythranthe guttata*, *Nicotiana tabacum* and *Rosa chinensis* using the complete protein sequences.** The neighbor-joining (NJ) tree was reconstructed using Clustal X 2.0 and MEGA 7.0 software with the pairwise deletion option. One thousand bootstrap replicates were used to assess the tree reliability. The three clusters of HmALMTs were distinguished by different colors.

qRT–PCR (Fig. 4). The expression patterns of *HmALMT* genes showed significant divergence among different tissues. None of the tested genes were expressed or downregulated in leaves. *HmALMT7/HmALMT10*, *HmALMT2/HmALMT3/HmALMT4/HmALMT5* and *HmALMT1/HmALMT9/HmALMT11* were uniquely expressed in roots, stems and flowers, respectively. Six *HmALMT* genes, *i.e.*, *HmALMT2*, *HmALMT3*,

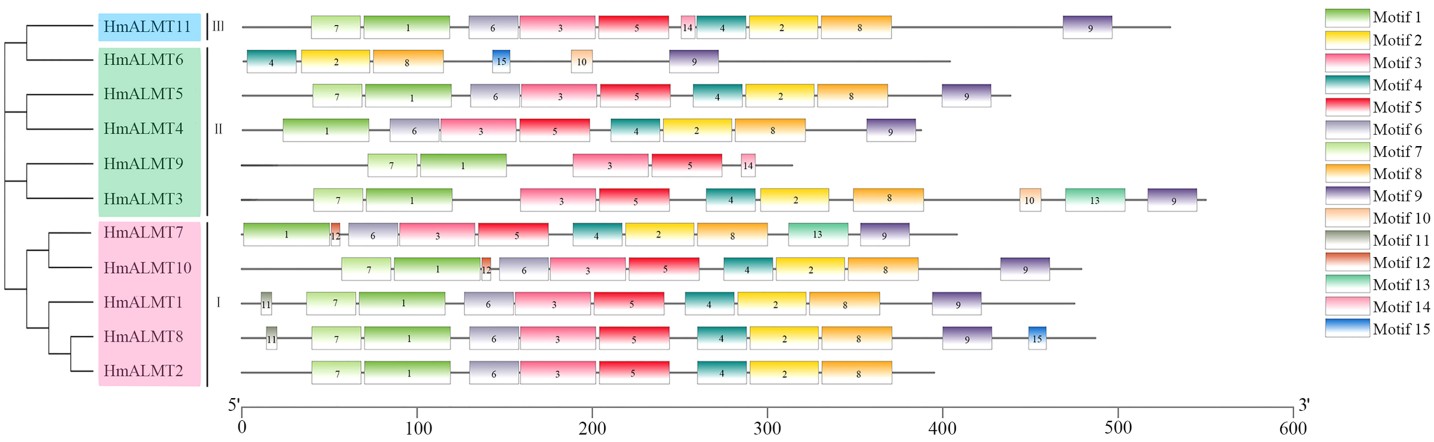

**Figure 2 The conserved motifs of hydrangea ALMTs according to phylogenetic relationship.** The phylogenetic tree was constructed by MEGA7.0 using the neighbor-joining method with 1,000 bootstrap replicates. The conserved motifs of the HmALMT proteins were detected using the online MEME program and drawn using TBtools software. Motifs 1–15 were displayed in different colored boxes. The sequence information of each motif was supplied in Table S2. The length of the protein could be estimated using the scale at the bottom.

**Table 2 The proportion of amino acids predicted by the secondary structure of HmALMT proteins.**

| Name | Alpha helix | Extended strand | Beta turn | Random coil |
|---|---|---|---|---|
| HmALMT1 | 287 (60.42%) | 53 (11.16%) | 12 (2.53%) | 123 (25.89%) |
| HmALMT2 | 250 (63.29%) | 51 (12.91%) | 12 (3.04%) | 82 (20.76%) |
| HmALMT3 | 302 (54.91%) | 56 (10.18%) | 11 (2.00%) | 181 (32.91%) |
| HmALMT4 | 254 (65.63%) | 44 (11.37%) | 13 (3.36%) | 76 (19.64%) |
| HmALMT5 | 269 (61.42%) | 54 (12.33%) | 15 (3.42%) | 100 (22.83%) |
| HmALMT6 | 197 (48.88%) | 40 (9.93%) | 10 (2.48%) | 156 (38.71%) |
| HmALMT7 | 240 (58.82%) | 39 (9.56%) | 14 (3.43%) | 115 (28.19%) |
| HmALMT8 | 281 (57.70%) | 58 (11.91%) | 11 (2.26%) | 137 (28.13%) |
| HmALMT9 | 180 (57.32%) | 48 (15.29%) | 8 (2.55%) | 78 (24.84%) |
| HmALMT10 | 290 (60.54%) | 53 (11.06%) | 17 (3.55%) | 119 (24.84%) |
| HmALMT11 | 305 (57.66%) | 49 (9.26%) | 11 (2.08%) | 164 (31.00%) |

*HmALMT4*, *HmALMT5*, *HmALMT6* and *HmALMT8*, were expressed at significantly higher levels in stems than in other tissues.

Under Al treatment, *HmALMT5* was significantly upregulated in all tissues. However, some genes were only upregulated in specific tissues after treatment with Al, including *HmALMT2*, *HmALMT10*, and *HmALMT3*, which were only upregulated in roots, stems, and leaves, respectively. Four genes were significantly upregulated in roots under Al stress, namely, *HmALMT2, HmALMT5, HmALMT6* and *HmALMT8*. Simultaneously, seven genes were significantly upregulated in stems, namely, *HmALMT1, HmALMT4, HmALMT5, HmALMT6, HmALMT9, HmALMT10* and *HmALMT11*. Furthermore, three genes were upregulated in leaves: *HmALMT3, HmALMT5* and *HmALMT11*. In addition, two genes were significantly upregulated in flowers, *HmALMT1* and *HmALMT11* (Fig. 5).

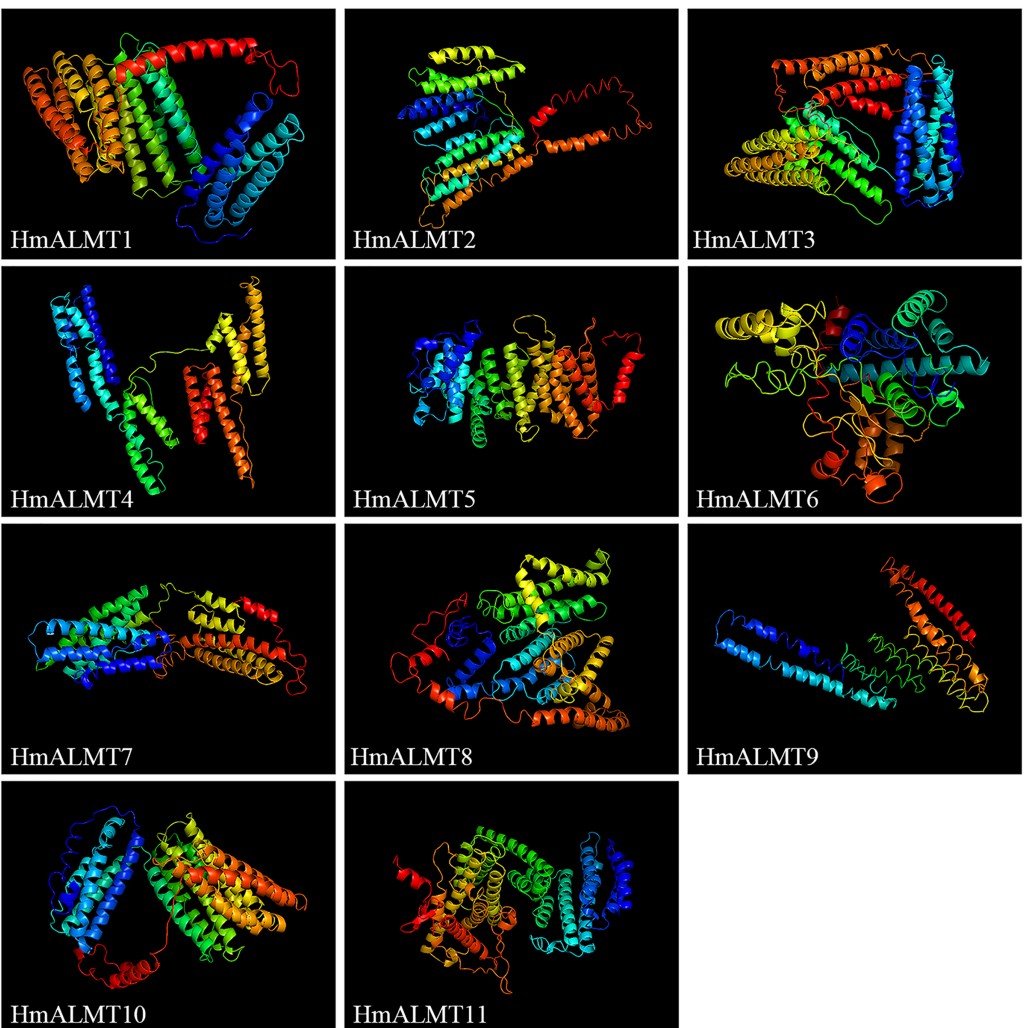

**Figure 3 Predicated 3D structures of HmALMT proteins.** Models were constructed by I-TASSSER online software and colored by rainbow from N- to C-terminus.

## *HmALMTs* expression enhanced Al tolerance in yeast

According to the qRT–PCR results, *HmALMT5*, *HmALMT9* and *HmALMT11* were selected for functional verification because of their strong induction in response to Al stress. All three yeasts overexpressing genes grew better than control (pYES2.0) on SG-U medium with 1 mM and 2 mM $Al_2(SO_4)_3$ (Fig. 6A). In liquid media supplemented with 1 mM $Al_2(SO_4)_3$, the growth of yeast cells expressing *HmALMT5*, *HmALMT9* and *HmALMT11* was higher than that of yeast cells expressing pYES2.0. Compared with yeast cells transformed with the empty vector cultured on the above medium, the yeast cells expressing *HmALMT9* showed significantly enhanced growth from the beginning to 120 h, while the yeast cells expressing *HmALMT5* and *HmALMT11* were stronger than the yeast cells transformed with the empty vector after 84 and 48 h, respectively (Fig. 6B).

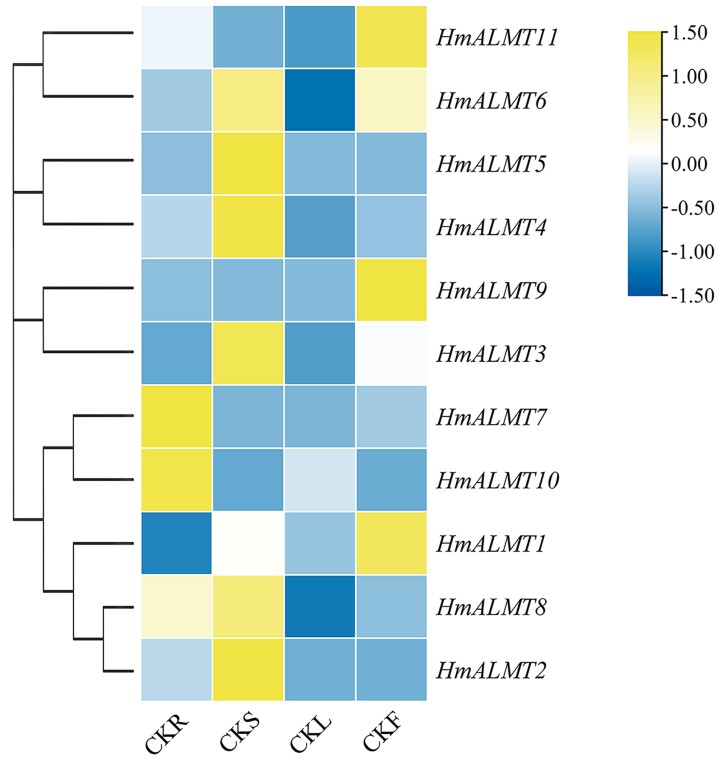

**Figure 4 Expression profiles of *HmALMTs* in different hydrangea tissues.** CKR, roots under control conditions; CKS, stems under control conditions; CKL, leaves under control conditions; CKF, flowers under control conditions. A heatmap was constructed based on relative expression levels. Different colors represent different expression levels, with red representing the highest value of gene expression.

## DISCUSSION

ALMTs play important roles in growth, development and response to Al stress in many plant species (*Collins et al., 2008*; *Hoekenga et al., 2006*). Previous researchers have cloned and characterized several ALMT homologs in many plants (*Angeli et al., 2013b*; *Ma et al., 2015*; *Yang et al., 2012*). However, the role of ALMT family members in hydrangea is still unclear. In this study, 11 ALMTs were identified from the transcriptome data of hydrangea. Phylogenetic analysis indicated that the *ALMT* gene family members in hydrangea can be divided into three large subfamilies, similar to the classification in *Arabidopsis* (*Kovermann et al., 2007*). Figure 1 shows that most HmALMTs were distributed in Cluster I and Cluster II, and few were distributed in Cluster III, which is similar to other species (*Ma et al., 2020*).

In HmALMTs, several highly conserved sequences were predicted, including TVVVVFE, AG(X)L, PW(X)(X)(X)Y, R(X)CA, K(X)G(X)(X)L(X)LVS, F(X)LTF, and WEP. All these sequences had significant functionality, as they were evolutionarily similar to their orthologs in different species. The highly conserved WEP motif (Trp Glu Pro) of HmALMTs was located on motif 4 (Fig. S1). Previous studies showed that ALMTs had a high similarity of secondary structure, with an N-terminal region containing five or six transmembrane domains and a long, strongly hydrophilic C-terminal being half length of

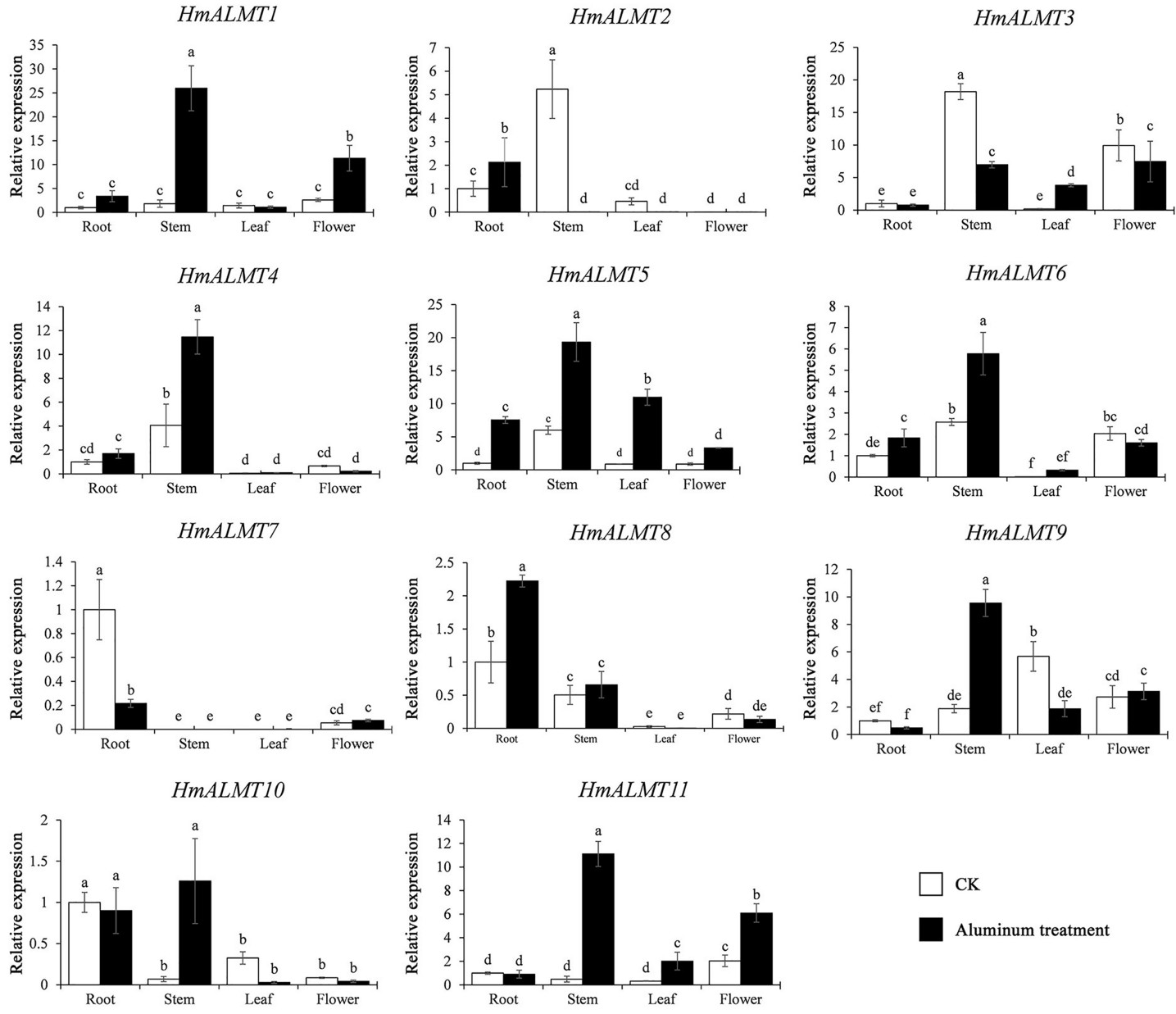

**Figure 5 Expression patterns of the 11 *HmALMTs* responding to Al treatment.** Total RNA was extracted from roots, stems, leaves, and flowers according to the manufacturer's instructions. The white columns represent the expression in different tissues of hydrangea under control conditions, and the black columns represent the expression in different tissues of hydrangea treated with Hoagland solution containing 15 mM $Al_2(SO_4)_3$. The relative expression of *HmALMTs* in different tissues was quantified by qRT–PCR. To calculate the relative expression level of each gene in different tissues, the transcript level in roots was used to normalize the transcript levels in other tissues. Different lowercase letters above the bars represent significant differences ($p < 0.05$) determined by Duncan's one-way analysis of variance. All primer sequences were listed in Table S1. The data were given as mean ± SD ($n = 3$).

the whole protein (*Dreyer et al., 2012*; *Ligaba et al., 2013*). A previous study indicated that all members of the ALMT family had a high structural similarity in the N-terminal half, whereas the C-terminal half had more variations (*Ligaba et al., 2013*). In hydrangea, HmALMTs were divided into N-terminal and C-terminal regions by WAG residues in motif 5 (*Brygoo et al., 2011*). Additionally, the N-terminus was composed of motifs 1, 3, 6

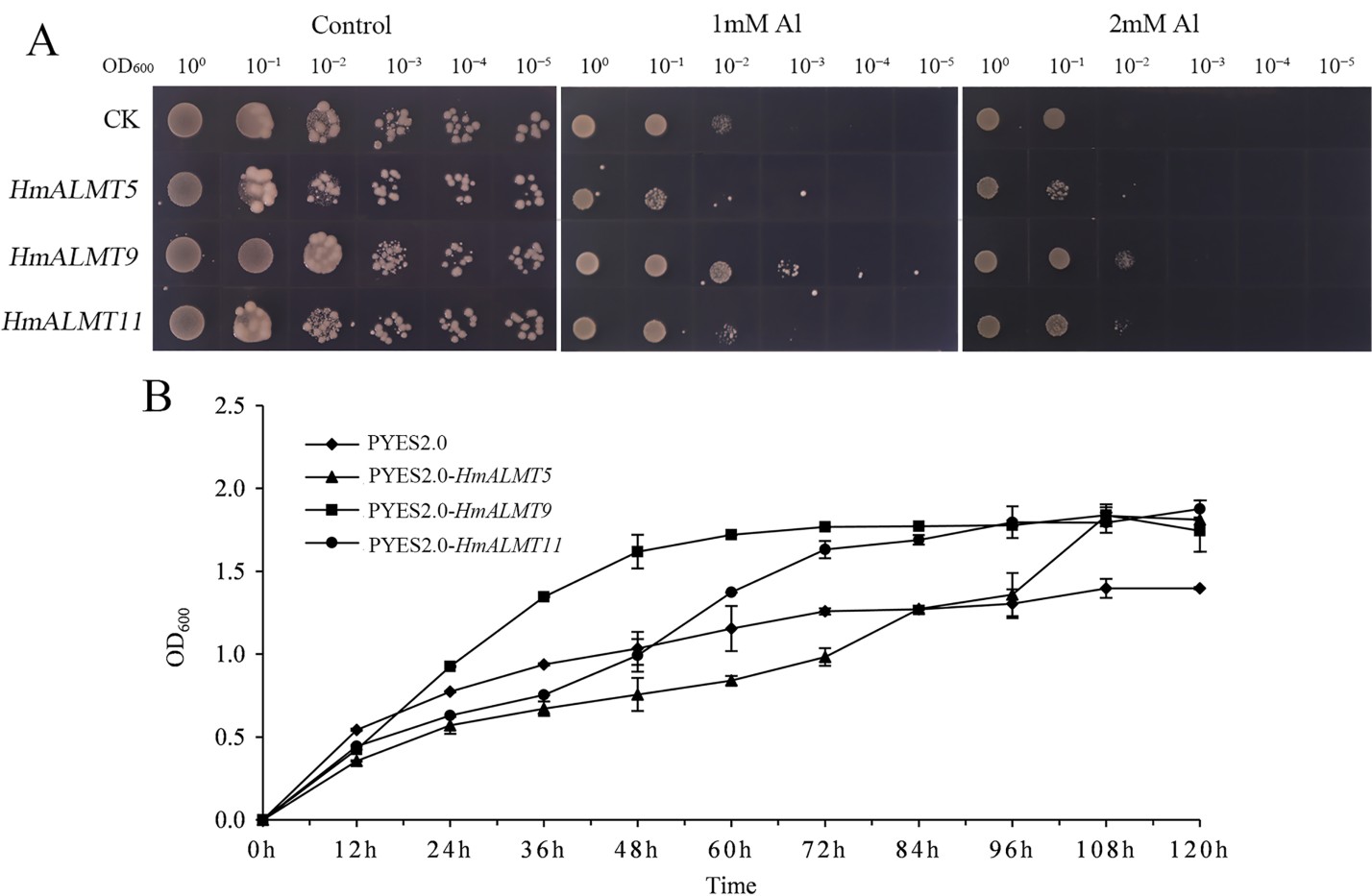

**Figure 6 Overexpression of *HmALMT5*, *HmALMT9* and *HmALMT11* increased the Al tolerance of Al in yeast.** (A) The growth of the BY4741 yeast mutant transformed with the empty vector pYES2.0 and pYES2.0 containing *HmALMT5*, *HmALMT9* and *HmALMT11*, respectively; (B) time-dependent growth of yeast strains in synthetic galactose uracil (SG-U) liquid medium supplemented with 1 μm $Al_2(SO_4)_3$.

and 7, which are ion transport channels that participate in $Al^{3+}$ signal transduction (*Brygoo et al., 2011*). *Furuichi et al. (2010)* identified three key residues (Glu274, Asp275 and Glu284) that eliminated Al-dependent transport changes after neutralization without affecting the basal transport activity. These residues were likely to be in the extracellular C-terminal region and act as a major determinant of $Al^{3+}$-activation (*Motoda et al., 2007*). The C-terminus of HmALMTs contains motifs 2, 4, 8 and 9 (rich in Glu274, Asp275 and Glu284), which are related to the regulation of the $Al^{3+}$ response. It was also found that glycine (G) and valine (V) were abundant in motif-1, which might be associated with protein dimerization. These results indicate that the high-level conservation of motifs demonstrates their value for structural integrity.

Numerous studies have shown that *ALMT* genes play critical roles in the Al stress response. In tomato (*Solanum lycopersicum*), higher activity of the *SlALMT9* promoter in the main root and lateral root was observed under $Al^{3+}$ treatment at low pH, which illustrates that *SlALMT9* could play a role in malic acid transport to detoxify Al (*Ye et al., 2017*). The overexpression of *TaALMT1* could significantly increase malate exudation and

Al resistance in wheat, barley (*Hordeum vulgare*) and tobacco (*Nicotiana tabacum*) suspension cells (*Delhaize et al., 2004*; *Pereira et al., 2010*). Rape *BnALMT1* and *BnALMT2* enhanced the resistance of transgenic tobacco cells to Al stress (*Ligaba et al., 2006*). Generally, malate excretion from plant roots is mediated by ALMT channels (*Meyer et al., 2010*). For example, *TaALMT1* (wheat), *AtALMT1* (*Arabidopsis*), *GmALMT1* (soybean), *ScALMT1* (rye), *MsALMT1* (*Medicago sativa*), *BoALMT1* (cabbage), and *HlALMT1* (*Holcus lanatus*) are involved in Al resistance by mediating malate secretion from the root tip (*Chen et al., 2012*; *Chen et al., 2013*; *Fontecha et al., 2007*; *Hoekenga et al., 2006*; *Liang et al., 2013*; *Ligaba et al., 2006*; *Zhang et al., 2017*). Here, a similar phenomenon was found in hydrangea. Under Al treatment, more than half of the genes were upregulated in all tissues. Heterologous expression of *HmALMTs* (*HmALMT5*, *HmALMT9* and *HmALMT11*) in yeast also conferred Al tolerance.

The expression pattern of *ALMTs* was tissue dependent. For example, in rice, *OsALMT1, -2*, and *-4* were expressed in leaves and roots, while *OsALMT7* and *-9* were only expressed in roots (*Liu et al., 2017*). A previous report showed that *HvALMT1* might contribute to nutrient delivery during grain development and germination and is expressed during grain development (*Xu et al., 2015*). In 34 soybean *GmALMT* members, 26 were upregulated in roots, leaves and flowers under phosphate starvation treatment (*Peng et al., 2018*). However, the ALMT family (*TaALMT1*, *BoALMT1*, *etc.*) mainly performed their functions in roots by adapting plants to Al toxicity (*Sasaki et al., 2004*; *Zhang et al., 2017*). *VvALMT9* was expressed in fruit and appears to play a role in the determination of fruit acidity in grape (*Vitis vinifera*) (*Angeli et al., 2013a*; Ma et al., 2015). In the present study, *HmALMT2, HmALMT5, HmALMT6* and *HmALMT8* were significantly upregulated in the roots compared with other tissues under Al stress, suggesting that the above four genes might be involved in the detoxification of Al (Fig. 5). Simultaneously, only one gene (*HmALMT2*) was uniquely upregulated in roots and downregulated in other tissues after Al treatment (Fig. 5). Therefore, it can be speculated that *HmALMT2* might control the malate acid exudation of hydrangea roots to cope with Al toxicity. Interestingly, *HmALMT10* was upregulated uniquely in stems, meaning that it may be involved in physiological processes such as malic acid transport. Malate acid is a key metabolite, and its function varies in different plant organelles (*Kochian, 1995*; *Wehr, Menzies & Blamey, 2003*). Therefore, the constitutive expression of *HmALMTs* could be expected to mediate the distribution of malate in plant tissues. These results indicate that ALMT family members may play different roles in response to Al toxicity in plants. In addition, these results suggest that the ALMT transporter genes are involved in the absorption, transportation and storage of aluminum ions in response to Al stress in roots and stems. In general, the diversity of expression patterns observed in *HmALMTs* strongly suggests that they support various functions of the whole hydrangea plant in response to Al tolerance.

## CONCLUSIONS

In conclusion, 11 ALMTs were identified in hydrangea, which can be divided into three clusters. According to the motif and structural analysis, HmALMTs are conserved during

amino evolution. The amino acids in the N-terminus mainly participate in $Al^{3+}$ signal transduction, while those in the C-terminus are primarily related to the regulation of the $Al^{3+}$ response. The qRT–PCR analysis showed different expression levels of *HmALMTs* in different tissues of hydrangea. The three selected *HmALMT* members showed transcriptional activation activities in yeast and positively regulated tolerance to Al stress. In conclusion, this study provides new guidance for the selection, cloning and functional analysis of *HmALMTs*, and reveals their important roles in regulation of Al tolerance and other physiological activities of hydrangea.

### Funding
This work was funded by the National Natural Science Foundation of China (No. 31901359). The funders had no role in study design, data collection and analysis, decision to publish, or preparation of the manuscript.

### Grant Disclosures
The following grant information was disclosed by the authors:
National Natural Science Foundation of China: 31901359.

### Competing Interests
The authors declare that they have no competing interests.

### Author Contributions
- Ziyi Qin conceived and designed the experiments, performed the experiments, analyzed the data, prepared figures and/or tables, authored or reviewed drafts of the article, and approved the final draft.
- Shuangshuang Chen conceived and designed the experiments, performed the experiments, analyzed the data, prepared figures and/or tables, authored or reviewed drafts of the article, and approved the final draft.
- Jing Feng analyzed the data, prepared figures and/or tables, and approved the final draft.
- Huijie Chen performed the experiments, prepared figures and/or tables, and approved the final draft.
- Xiangyu Qi analyzed the data, prepared figures and/or tables, and approved the final draft.
- Huadi Wang performed the experiments, prepared figures and/or tables, and approved the final draft.
- Yanming Deng conceived and designed the experiments, authored or reviewed drafts of the article, and approved the final draft.

### Data Availability
The raw data from qRT-PCR and OD values measured during yeast growth, reflecting the tissue expression and overexpression of HmALMT5, -9, -11 in yeast, are available in the Supplemental Files.

## Supplemental Information

Supplemental information for this article can be found online at http://dx.doi.org/10.7717/peerj.13620#supplemental-information.

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
