# Peer review of "Identification of aluminum-activated malate transporters (ALMT) family genes in hydrangea and functional characterization of HmALMT5/9/11 under aluminum stress"

_PeerJ, doi:10.7717/peerj.13620_

## Round 0.1 · original submission · Major Revisions

· Academic Editor

Major Revisions

Please take the comments from two reviewers into consideration for revision.

Reviewer 1 ·

Basic reporting

I think the Figure legend is too simple, some key points should be given, e.g. in Figure5, the Al concentration and how the treatment was conducted.

Experimental design

no comment

Validity of the findings

no comment

Additional comments

The manuscript by Qin and colleagues describes 11 ALMTs from Hydrangea, their physicochemical properties, phylogeny, conserved motifs and protein structure were analyzed. Moreover, three members were selected to study their Al tolerance in yeast. To some extent, these results help us to improve recognition of ALMTs from Hydrangea.
The manuscript could be improved by addressing the following points:
Major Concerns:
In M&M, lines 106-109,
1) Why 15 mM Al2(SO4)3 was used? Generally, the plant growth will be inhibited at micromolar level of Al3+. So, the author should cite related literature or give the reasons.
2) Al3+ will complex with some anions (e.g. PO4, OH-, SO4, EDTA…) in Hoagland’s nutrient solution at pH 4.5. What is the Al ion activity in Hoagland solution? The author should calculate it by GEOCHEM-EZ.
3) Line 229-230: HmALMT7 was up-regulated in flowers? The error bar is too large for HmALMT7 expression in flowers (also HmALMT9 in leaf, HmALMT8 in CK root). The qRT-PCR experiments (HmALMT7, HmALMT8, and HmALMT8 expressions in Figure5) should be redone. I also suggest the author makes an analysis of variance between CK and aluminum stress (Figure 5).

Minor Concerns:
1) Line31, ‘tissue specific’ should be ‘tissue-specific’.
2) About the writing format of ‘ALMT gene family’, sometimes ‘ALMT gene family’ (e.g. line 35, line 67, line 72, line92, line213…), sometimes ‘ALMT gene family’ (e.g. line90), please check the whole MS.
3) Lines 66: the sentence should be ‘the secretion of malic acid and citric acid was controlled by aluminum-activated malate transporter (ALMT) and multidrug and toxic compound extrusion transporter (MATE), respectively.’
4) Line73: ‘scALMT1’ should be ‘ScALMT1’.
5) Line 188: only one gene (HmALMT11) in Cluster III…, the word ‘gene’ should be omitted.
6) Line212: …under Al Stress, ‘Stress’ should be ‘stress’.
7) Line214-215: ‘…leaves, stems and flowers harvested from the control and treated plants’ (refer to Figure 4). What treatment has been done?
8) In Figure 5, in order to be consistent with the whole manuscript, I suggest the author replaces aluminium with aluminum.
9) Line281: The right parenthesis symbol is missing.
10) Lines 287-288: HmALMT2 was significantly up-regulated in the roots than other tissues? According to Figure4, HmALMT2 was significantly up-regulated in the stem rather than roots, please check.
11) Lines 290-291: only one gene (HmALMT2) was uniquely up-regulated in roots and down-regulated in other tissues after Al treatment (see Figure 4). Figure4 or Figure5?
12) Line 303: ‘In a conclusion’ should be ‘In conclusion’.
13) The file named ‘Supplemental_Table_S3’, however, the title in this table is ‘Table S2 The information of 3D structure prediction’. Please check.

Reviewer 2 ·

Basic reporting

In the manuscript entitled “Identification of Aluminum-Activated Malate Transporters (ALMT) family genes in Hydrangea and functional characterization of HmALMT5/9/11 under Al stress” (#71046), the authors identified 11 candidate ALMT family members from the transcriptome data in hydrangea. The amount of data analyzed is large and the results obtained are abundant.

I believe that this article can meet the requirements published in PeerJ after major reversion.

Experimental design

well

Validity of the findings

no comment

Additional comments

1. The organization of paper writing needs to be further improved.
2. Some normative writing needs attention, such as Table 1: PI should be pI, and italic.
3. Line 56, Reference formats need to be proofread, e.g. Bernhard Wehr et al. 2003; Line 70, 70 Ligaba et al. 2006; Ligaba et al. 2012 should be Ligaba et al. 2006; 2012.
4. The conditions for plant growth are not well defined, and need to be further defined.
5. Figure 6 lacks statistical analysis.
6. Overall, major text in the discussion looks fine. However, a better discussion will help improve the manuscript.
7. Reference needs further examination. Retain the relevant and recent literature/report.
8. Before being considered for publication, the manuscript will be corrected by a fluent English speaker, alternatively please use one of the commercial English language editing services available.

---

## Round 0.2 · Minor Revisions

· Academic Editor

Minor Revisions

Please make corrections based on the comments from the reviewers.

Reviewer 1 ·

Basic reporting

no comment

Experimental design

no comment

Validity of the findings

no comment

Additional comments

In this Ms, the authors have answers all questions carefully. However, some questions should be addressed:
Minor Concerns:
1) In M&M, why the plant Al treatment was conducted in soil with 15 mM Al2(SO4)3 irrigation rather than hydroponic culture? Generally, the hydroponic conditions are stable and easy to control.

2) In your response, the final pH of Hoagland solution after adding 15mM Al2(SO4)3 is between 2.5-3. The solution is strong acid. In the MS, Line 106-107, Hoagland solution (500 ml per pot) with or without 15 mM Al2(SO4)3 was used to irrigate…, Line 214-215, the control [Hoagland solution without Al2(SO4)3]…. My question is the pH of the control treatment? 2.5-3? 4.0-4.5 (usually used for Al treatment)? or normal pH (~5.8)? The author should indicate this in M&M part. Similar question is presented in Line 234, …grew better than control (pYES2.0) on SG-U medium. The possible effect of pH on the phenotype/growth should be excluded.

3) I note that the expression of HmALMT8 in Al-treated root is nearly 8-fold compared with CK in previous version, it markedly declines (less than 2.5-fold) in the revised version (Figure 5). It varies greatly between the two versions. What are the possible reasons?

4) Line 226, Line 289 OsALMT7 and -9, Line 297, the ‘and’ should not be italic.

5) In reference lists, line 327, the ‘Vitis vinifera’ should be italic.

6) Table 1 (including the Note below the table): PI should be pI, and italic.

Reviewer 2 ·

Basic reporting

no comment

Experimental design

no comment

Validity of the findings

no comment

Additional comments

no comment

---

## Round 0.3 · accepted · Accept

· Academic Editor

Accept

Based on two reviewers' recommendations, your manuscript is deemed to be acceptable.

Reviewer 1 ·

Basic reporting

no comment

Experimental design

no comment

Validity of the findings

no comment

Additional comments

The authors answer all questions I concerned very well. I think the manuscript should be accepted for publication in PeerJ.